# Branched Sulfonimide-Based Proton Exchange Polymer Membranes from Poly(Phenylenebenzopheneone)s for Fuel Cell Applications

**DOI:** 10.3390/membranes11030168

**Published:** 2021-02-27

**Authors:** Sabuj Chandra Sutradhar, Sujin Yoon, Taewook Ryu, Lei Jin, Wei Zhang, Whangi Kim, Hohyoun Jang

**Affiliations:** 1Department of Applied Chemistry, Konkuk University, Chungju 27478, Korea; sabujchandra@gmail.com (S.C.S.); ysj920126@naver.com (S.Y.); gundam0924@naver.com (T.R.); jinlei8761@naver.com (L.J.); arno_zw@hotmail.com (W.Z.); wgkim@kku.ac.kr (W.K.); 2Department of Liberal Art, Konkuk University, Chungju 27478, Korea

**Keywords:** proton exchange membrane, sulfonimide, nickel catalyzed polymerization, ion exchange capacity, dimensional stability, proton conductivity

## Abstract

Improved proton conductivity and high durability are now a high concern for proton exchange membranes (PEMs). Therefore, highly proton conductive PEMs have been synthesized from branched sulfonimide-based poly(phenylenebenzophenone) (SI-branched PPBP) with excellent thermal and chemical stability. The branched polyphenylene-based carbon-carbon backbones of the SI-branched PPBP membranes were attained from the 1,4-dichloro-2,5-diphenylenebenzophenone (PBP) monomer using 1,3,5-trichlorobenzene as a branching agent (0.1%) via the Ni-Zn catalyzed C-C coupling reaction. The as-synthesized SI-branched PPBP membranes showed 1.00~1.86 meq./g ion exchange capacity (IEC) with unique dimensional stability. The sulfonimide groups of the SI-branched PPBP membranes had improved proton conductivity (75.9–121.88 mS/cm) compared to Nafion 117 (84.74 mS/cm). Oxidation stability by thermogravimetric analysis (TGA) and Fenton’s test study confirmed the significant properties of the SI-branched PPBP membranes. Additionally, a very distinct microphase separation between the hydrophobic and hydrophilic moieties was observed using atomic force microscopic (AFM) analysis. The properties of the synthesized SI-branched PPBP membranes demonstrate their viability as an alternative PEM material.

## 1. Introduction

Proton exchange membrane fuel cells (PEMFCs) have become the most popular renewable energy resource because of their high efficiency, high energy density, quiet operation, and environmental friendliness [1,2,3,4,5]. Proton exchange membranes (PEMs) are the key component of these fuel cells and are primarily based on a perfluorosulfonic acid called Nafion. Despite Nafion’s high proton conductivity and good chemical and oxidative stability, its low-temperature operation, high manufacturing cost, harsh chemical modifications and high fuel permeability have stimulated researchers to find alternatives to Nafion [6,7,8,9,10]. Therefore, aromatic hydrocarbon-based polymer membranes, such as sulfonated poly(ether ether ketone)s (PEEK) [3,11,12,13,14], polysulfones (PSF) [15,16,17] and poly(arylene ether)s (PAE) [18,19,20,21] have been studied for over a decade as alternative membranes to Nafion. A class of composite polymer electrolyte membranes containing SPSU-LDH [22], PBI-SiO_2_ [23], and SGO-SPAES [24] have shown interconnecting proton transport channels, a phase-separated morphology and enhanced thermo-mechanical resistance, water retention capacity and high dimensional stability. However, the lack of considerable thermal and oxidative stability made the use of these polymer membranes as proton exchange membranes inappropriate. Meanwhile, to improve the oxidative stability of these linear polymers, researchers have suggested using cross-linked polymers for fuel cell applications [25,26,27]. However, the insolubility of these cross-linked membranes in common organic solvents has impeded their ability to used as PEMs in commercial processes. Researchers have also suggested that branching polymer membranes, which have good solubility in organic solvents, have an excellent potential to be used as PEMs materials. However, very few branched PEMs have been studied compared to cross-linked PEMs [4,28,29,30,31,32,33,34,35,36]. Hay et al. [4] has reported branched polymers containing sulfonic acid as end groups whereas Ueda et al. [34] has studied sulfonated block copolymer with side-arms. Park et al. [37]. and Wang et al. [29,38] have reported sulfonated block copolymers containing 0.4% and 2–4% branching agents for getting high molecular weight. Comb-shaped sulfonated polymers [39,40,41,42,43,44] have also been reported to have highly branched backbones with flexible side chains. These comb-shaped sulfonated polymers show good mechanical properties. Nevertheless, most of these branched polymer membranes are susceptible to nucleophile attack as the polymer backbones are based on ether linkages, and acid functional groups are attached to the main chain [17,45]. Conversely, polyphenylene membranes exhibit excellent durability, thermo-oxidative stability and good solubility in organic solvents [46,47,48,49,50,51]. Additionally, sulfonimide acid functional groups have super gas-phase acidity [49]. The pendant side-chain type sulfonimide groups provide very distinct micro-phase morphology. Moreover, the fluorine atom of the sulfonimide acid moiety enhances the chemical stability of the polymer membranes by protecting the polymer backbones from free radical attacks during fuel cell operation.

Therefore, in this study we have synthesized polyphenylene membranes without any ether linkages using Ni-Zn catalyzed C-C coupling polymerization in order to overcome the thermal, chemical stability and proton conductivity obstacles associated with polymer membranes. The as-synthesized polyphenylenes backbones containing pendant benzoyl groups are believed to exhibit excellent thermal and chemical stability. Furthermore, super gas-phase acidic fluoro-sulfonimide acid functional groups have been introduced in the polymer side chains to improve proton conductivity, chemical stability, and structural flexibility.

## 2. Materials and Methods

### 2.1. Materials

2,5-dichloro-p-xylene, potassium permanganate, benzene, pyridine, nickel bromide, thionyl chloride, triphenylphosphine and zinc powder were purchased from TCI (Tokyo, Japan), Sigma-Aldrich (St. Louis, MO, USA) and Alfa Aesar (Ward Hill, MA, USA). Chlorosulfonyl isocyanate was supplied by Chumbuk (Korea). Chlorosulfuric acid, formic acid, and antimony trifluoride, respectively, were purchased from Junsei (Tokyo, Japan), Daejung Chemicals (Busan, Korea), and Alfa Aesar. Commercial solvents such as dimethylacetamide (DMAc), dimethyl sulfoxide (DMSO), dichloromethane (DMC), chloroform, methanol, ethanol and acetone were purchased from Sigma-Aldrich and used without further purification.

### 2.2. Synthesis of Branched Poly(phenylenebenzophenone) (PPBP)

Preparation of the reagents and catalysts was carried out in a glove box under an N_2_ condition. NiBr_2_ (0.174 g, 0.79 mmol), Zn powder (3.03 g, 48.7 mmol) and triphenylphosphine (1.67 g, 6.4 mmol) were put into a three-neck flask. Additionally, 1,4-dichloro-2,5-diphenylenebenzophenone (PBP) (2.0 g, 7.9 mmol) and 1,3,5-trichlorobenzene (0.002g, 0.1% of PBP) were put into a one-neck flask. To begin the reaction, DMAc (3–5 mL) was added into the catalytic flask by a syringe and fitted with a mechanical stirrer under a flowing nitrogen system. It was stirred gently at 80 °C until it turned blood-red. Thereafter, the PBP monomer was dissolved into DMAc solvent and added into the catalytic flask through a syringe. The resultant mixture was stirred at 100 °C until it became a viscous, jelly-like, mixture. The viscous mixture was diluted with 8–10 mL DMAc and cooled to room temperature before precipitation. Then, the solution was poured while stirring into distilled water containing 30% HCl. When solids formed, they were collected through filtration and washed first with distilled water and then with acetone. Finally, the polymer (PPBP) was dried for 12 h in a vacuum oven at 60 °C.

### 2.3. Sulfonation of the Branched Polymer (S-Branched PPBP)

PPBP branched polymers (1.5 g) were dissolved in dichloromethane (10 mL) at 0 °C before slowly being added to concentrated chlorosulfuric acid (3.18 mL, 47.8 mmol) and stirred for 6 h at 80 °C. The black-colored solution was then poured into distilled ice water and washed to remove residual acid. Finally, the solid polymer was rinsed with distilled water on a filter and dried at 60 °C for 24 h.

### 2.4. Conversion into Sulfonimide form of the Sulfonated Branched Polymers (SI-Branched PPBP)

Initially, S-branched PPBP (1.0 g) was dispersed in 10 mL tetrachloroethane (TCE) and thionyl chloride (20 mL). Then, the mixture was refluxed at 75 °C for 24 h by adding a few drops of DMF. The resultant mixture was evaporated to obtain a paste-type residue. After dissolving again in dichloromethane (30 mL), fluoro-sulfonimide (0.8 g, 8 mmol) was added slowly by a syringe at 0 °C. The resultant mixture was stirred at room temperature until the solution color turned a deep orange. Finally, the mixture was carefully poured into a mixture of methanol and distilled water (5:5) to collect the sulfonimide grafted branched PPBP polymers (SI-branched PPBP) and dried at 60 °C.

### 2.5. Characterizations and Measurement of Membranes Properties

Membranes with a thickness of 25 µm were made by dissolving 3% *w/v* SI-branched PPBP polymers in the DMAc solvent and exposing them overnight under an IR lamp on a glass plate. Typically, a RheoSense hts-VROC™ viscometer was used to measure the viscosity of the SI-PPBP polymers. The structural properties of the synthesized DCBP monomer and SI-PPBP polymers were studied using JEOL (400 YH) for ^1^H, ^19^F-NMR and FT-IR spectra with Nicolet iS5 FT-IR Spectrometry (Serial no. ASB1100426). The constant weighted polymer membranes were immersed into distilled water for 24 h at 80 °C and measured water uptake was as follows:Water uptake, WU (%) = {(W_wet_ − W_dry_)/W_dry_} × 100%(1)
where W_dry_ and W_wet_ correspond to the weight of the membranes in dried and wet conditions, respectively.

Next, the SI-branched PPBP membranes were stirred into a 1N NaCl solution for 24 h at 80 °C to exchange the H^+^ ions with Na^+^_._ Subsequently, the exchanged H^+^ was evaluated by titration with 0.01N NaOH solution to measure the ion exchange capacity (IEC) as follows:IEC (meq./g) = (V_NaOH_ × M_NaOH_)/W_dry_ membrane(2)
where V_NaOH_, M_NaOH_ and W_dry_ correspond to the volume, molarity of the NaOH, and weight of the membrane, respectively.

Consequently, other membrane properties, i.e., hydration number (λ) and dimensional changes, were also evaluated as follows:Λ = (10 × WU%)/(IEC × 18)(3)
Δl (%) = {(l_wet_ − l_dry_)/l_dry_} × 100(4)
Δt (%) = {(t_we_t − t_dry_)/t_dry_} × 100(5)
where, λ, l and t represent the hydration number, length and thickness of the membranes, respectively.

The through-plane conductivity of the SI-PPBP membranes was conducted using the MTS 740 membrane test system (Scribner Associates Inc., Southern Pines, NC, USA) with a Newton 4th Ltd. (N4L) impedance analysis interface (PSM 1735). Constant alternating current was applied through both electrodes with the membranes in the middle. A specific humidity (30–90%) and temperature (30–90 °C) was maintained during the operation. The conductivity of the membranes was evaluated following the equation:σ = [L/(R_mem_ × A)](6)
where L, A and R_mem_ denote the thickness, electrode area, and corresponding resistance of the membranes, respectively.

The thermal property of the SI-PPBP polymer membranes was investigated with a TGA-N 1000 analyzer (Scinco, Chicago, IL, USA) at 30–800 °C with a scan rate of 20 °C/min under air conditions.

Additionally, the chemical stability of the membranes was evaluated by heating the membranes into Fenton’s reagent (3 ppm Fe^2+^_,_ 3% H_2_O_2_) at 80 °C and recording the chemical degradation by measuring the weight of the membranes at 1 h time intervals for 9 h.

The hydrophilic and hydrophobic phase separation of the membrane was assessed using trapping mode atomic force microscopy (AFM) with a Nanoscope (R) IIIA and microfabricated cantilevers with an amplitude setpoint of 0.7785 V.

Membrane electrode assemblies (MEAs) with an active area of 25 cm^2^ were prepared using a decal method based on a catalyst coated membrane (CCM). A 20 wt% wet-proofed Toray carbon paper (TGPH-060, Toray Inc.) of 190 mm thickness was employed as a gas diffusion layer (GDL) for the anode and cathode sides. Carbon-supported Pt (Hispec 13100, Johnson Matthey Inc.) was used as a catalyst for both anode and cathode and the loading of the catalyst layer was 0.29 mg Pt/cm^2^. Immediately afterward, the catalyst layer was transferred onto the membrane at 120 °C and 10 MPa for 5 min by decal method to make the CCM. The GDL was placed on the anode and cathode side of the CCM to form the MEAs. After assembling the single cell, the MEAs were fully hydrated by feeding fully humidified N_2_ into the single cell for 2 h. During the operation, fully humidified H_2_ and air at 70 °C were fed into the anode and cathode, respectively. The stoichiometry of hydrogen to air was maintained to be 1.5/2.0 and the relative humidity 100/100%. After the activation procedure, polarization curves were measured with a commercial test station (Scitech, Korea Inc) at a temperature of 70 °C and at ambient pressure. Polarization measurements were started at the OCV(Open Circuit Voltage) and the cell was operated in the galvanostatic mode with a scan rate of 36 mA/s for each step.

The tensile stress-strain properties of the membranes were tested using a Com-Ten Industries 95T series load frame equipped with a 200 lbf load cell and computerized data acquisition software. Samples of 9 mm width were deformed at a crosshead speed of 5 mm/min with a gauge length of 30 mm.

## 3. Results and Discussion

### 3.1. Preparation of the Monomer

PBP monomer was produced from 2,5-dichloro-p-xylene to subsequent oxidation, chlorination, and Friedel-Crafts acylation reaction with manganese (IV) oxide, thionyl chloride and benzene (Scheme S1). ^1^H NMR data were in close agreement with our previous work [47]. All the phenyl protons appeared at 7.47–7.88 ppm. The protons in the ortho position (H_a_) to chlorine atoms appeared at an upfield near 7.48–7.49 ppm because of the mesomeric effect of the chlorine atoms (Appendix A). Additionally, the proton peaks for the ortho, para and meta position of side phenyl rings shifted to the downfield as a result of the electron-withdrawing effect of the carbonyl group and appeared at near 7.83–7.88 (H_b_), 7.64–7.69 (H_d_) and 7.50–7.56 (H_c_) ppm, respectively.

### 3.2. Preparation of the Sulfonimide Branched PPBP Polymers (SI-Branched PPBP)

The SI-branched PPBPs were synthesized from PBP monomer and branching agent 1,3,5-trichlorobenzene via a Ni-Zn catalyzed C-C coupling reaction proceeding addition and elimination (Scheme 1). The high molecular weight PPBP polymer was applied to the specific mole ratio of the catalysts (NiBr_2_: PPh_3_: Zn = 1:8:60) [51,52,53,54]. Chlorosulfuric acid was used for the sulfonation of the branched PPBP polymer. Thereafter, all the sulfonic acid groups of the S-PPBP were converted to SI-branched PPBP by a sequential reaction with SOCl_2_ and sulfamoyl fluoride (FSO_2_NH_2_) (Scheme 1).

However, ^1^H NMR, ^19^F NMR and FT-IR spectroscopy were used to characterize the chemical structure of the synthesized polymers ( Figure 1; Figure 2, respectively). Figure 1 shows that all phenyl protons for the branched PPBP and S-branched PPBP polymers appear at 6.30–7.79 ppm. Noticeably, the –SO_3_H peaks for branched S-PPBP polymers were found in the upfield at 3.90–4.70 ppm as it conjugated with solvent DMSO and water (Figure 1b). However, after being converted into sulfonimide form, the phenyl protons were observed at 6.80–8.20 ppm slightly shifted downfield by the sulfonimide groups (Figure 1c). The delocalization of the sulfonimide anion is caused by these chemical shifts. Markedly, the –NH– protons appear at 5.20–5.90 ppm as lumpy peaks. On the other hand, the conversion of sulfonimide groups confirms the disappearance of the protons of sulfonic acid groups at 3.90–4.70 ppm. Furthermore, in the ^19^F-NMR, a notable peak of the fluorine atom in SI-branched PPBP polymer shows at −139.44 ppm and successfully attached to the sulfonimide groups on the S-PPBP polymer (Figure 1d).

Figure 2 shows the FT-IR spectra for the synthesized polymers. The broad stretching frequencies ranging from 2800–3650 cm^−1^ correspond to the –OH groups of the S-branched PPBP polymers. The asymmetric and symmetric stretching vibration of sulfone units (S=O) in sulfonate groups appear at 1420 and 1140 cm^−1^ and the stretching frequencies of carbonate groups (C=O) appear at 1670 cm^−1^. However, the sharp characteristic peaks for N–H stretching found at 3500 cm^−1^ and N–H bending were overlapped with a C=O stretching vibration at 1610 cm^−1^ for the SI-branched PPBP polymers. Additionally, the frequencies at 650 cm^−^^1^ for S–F stretching, with some contribution from S–N–S angle bending, confirms that the sulfonimides were incorporated into the sulfuric acid groups.

### 3.3. IEC, Water Uptake and Dimensional Stability of Membranes

Proton conductivity depends on ion exchange capacity (IEC) values i.e., the content of the sulfonimide functionalized groups of the membranes. The synthesized SI-branched PPBP membranes show a gradual increase in water uptake as IEC values increase. (Figure 3). The measured water uptake for the SI-branched PPBP membranes ranged from 25.3% to 66.8%, whereas Nafion 117^®^ typically shows 32.17% water uptake.

Dimensional stabilities (through-plane (Δt) and in-plane (Δl) changes) are important factors for membrane stacks which are usually affected by temperature and humidity. The SI-branched PPBP membranes exhibited lower Δt (3.1%, 4.7%, and 7.6%) and Δl (5.2%, 6.3%, and 10.8%) values compared to Nafion 117^®^ (Δt = 32.21 and Δl = 14.10%) (Table 1). The wholly aromatic and branched polymer backbones of the SI-branched PPBP membranes are the main cause of this lower-dimensional change.

### 3.4. Proton Conductivity of the SI-Branched PPBP Membranes

Measurements of the proton conductivity of the synthesized membranes were carried out at a range of temperatures (30–90 °C) and humidities (30–90% RH). In Figure 4a, the proton conductivities are demonstrated at a specified temperature (RT = 90 °C) with different relative humidities (30–90% RH). A gradual increase in proton conductivity for the SI-branched PPBP membranes was observed throughout the applied humidity range. Noticeably, SI-branched PPBP-40 membranes exhibited higher conductivity than Nafion 117^®^ over 80% relative humidity_._ The delocalization of the sulfonimide groups within the polymer network retains and facilitates the high proton transfer capability of the SI-branched PPBP membranes. Additionally, Figure 4b displays the proton conductivity of the SI-branched PPBP membranes at gradually increasing temperatures (30–90 °C) with a fixed relative humidity (RH = 90%). Markedly, the SI-branched PPBP-40 membrane shows higher proton conductivity (121.88 mS/cm) than Nafion 117^®^ (84.74 mS/cm) at the same condition. Moreover, synthesized SI-branched PPBP-20 and 30 membranes also exhibited almost analogous conductivities (75.9, 83.9 mS/cm respectively) to Nafion 117^®^ at the 90 °C and 90% RH condition. The higher super acidity and the greater number of sulfonimide groups that formed continuous ionic channels in the SI-branched PPBP-40 membranes typically accounted for the higher proton conductivity.

### 3.5. Thermo-Oxidative Stability of Membranes

Figure 5 shows the thermal degradation curves for the thermal resistance of the membranes in an air atmosphere. The branched PPBP polymer exhibited weight loss at 350 °C for the polymer backbone degradation. In the case of SI-branched PPBP membranes, the degradation showed two-step weight loss at 225–320 °C and 350–450 °C, which corresponded to the decomposition of sulfonimide acid groups and degradation of the polymer main chains, respectively. Therefore, the exhibited thermal stability of the SI-branched PPBP copolymers is suitable for fuel cell applications.

### 3.6. Chemical Stability of Membranes

Chemical stability was measured by free radical degradation of the SI-branched PPBP membranes and studied using Fenton’s reagent. Figure 6 shows that the SI-branched PPBP membranes have superior chemical stability compared to SPAES-40 membranes against free radical attack. The excellent chemical stability of the SI-branched PPBP membranes is due to a lower susceptibility of their C-C bonded polymer backbones to free radical attack than membranes containing ether linkages (SPAES-40). Moreover, the fluorine atoms of the sulfonimide groups resist free radical attacks. Therefore, the SI-branched PPBP membranes can provide long-term stability in fuel cell applications.

### 3.7. Morphology of the Membranes

Figure 7 shows the bare eye views for branched PPBP (a, b and c) and morphological images for the SI-branched PPBP polymer membranes (e, f and g) where the bright and dark colored portions in e, f and g represent the hydrophobic and hydrophilic domain, respectively [55]. The location of the sulfonimide groups in the pendant phenyl rings provides a clear phase-separation between hydrophobic and hydrophilic domains for the SI-branched PPBP membranes. In the AFM images, the SI-branched PPBP-40 membranes exhibit a great number of black dark spots as an adequate number of ion-conducting sulfonimide groups made a continuous channel for proton passage throughout the polymer network. On the contrary, SI-branched PPBP-20 membranes exhibited very few dark spots, indicating interrupted ionic channels due to the presence of a smaller quantity of sulfonimide groups.

### 3.8. Cell Performance of the SI-Branched PPBP Membranes

The single-cell performance of the SI-branched PPBP membranes is measured in terms of cell voltage and power density (Figure 8). Markedly, the SI-branched PPBP-40 membranes have a power density from 0.49 to 0.63 W/cm^2^ as compared to Nafion 117^®^ 0.62 W/cm^2^. However, synthesized SI-branched PPBP-20 and 30 membranes also showed comparable maximum power densities of 0.49 and 0.56 W/cm^2^, respectively.

## 4. Conclusions

A series of branched sulfonimide-based poly(phenylenebenzophenone)s membranes (SI-branched PPBP) have been successfully synthesized from 1,4-dichloro-2,5-diphenylenebenzophenone (PBP) monomers using 1,3,5-trichlorobenzene as a branching agent (0.1%) by Ni–Zn catalyzed C–C coupling polymerization. The nickel-zinc catalyzed C–C coupled polymer backbones with branching agent improved the thermal and chemical properties of the synthesized SI-branched PPBP membranes. Moreover, sulfonimide groups had significantly improved thermal properties because of C–C bonded polymer backbones and improved chemical stability because of a low susceptibility to free radical attack. The SI-branched PPBP-40 membranes showed high proton conductivity of 121.88 mS/cm at 90 °C/90% RH, which is almost 43% higher than Nafion 117^®^ (84.74 mS/cm). The excellent water retention ability of the SI-branched PPBP membranes improved electrochemical performances in single-cell PEMFCs under 70 °C/100% RH, reaching a peak power density value of 0.631 W/cm^2^ @ 70 °C/100% RH. The thermogravimetric analysis and Fenton’s test results also demonstrated the excellent thermal and chemical properties of the synthesized SI-branched PPBP copolymers. Additionally, the pendant sulfonimide groups allowed for a microphase separation between the hydrophilic and hydrophobic segments which enhanced proton conductivity. Therefore, the synthesized SI-branched PPBP membranes could be inexpensive electrolyte membranes for PEMFCs.

## Data Availability

Data sharing is not applicable to this article.

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
