# Peer review of "Branched Sulfonimide-Based Proton Exchange Polymer Membranes from Poly(Phenylenebenzopheneone)s for Fuel Cell Applications"

_membranes, 2021, doi:10.3390/membranes11030168_

Round 1
Reviewer 1 Report
The manuscript reports the preparation of branched sulfonimide-based poly(phenylenebenzophenone) proton exchange membranes. The results are interesting but the work should be improved before publication:
- In PEMFCs, the mechanical properties are also important because the membranes must tolerate severe mechanical solicitations. Please show the stress-strain tests on the proposed membranes.
- The development of sulfonimide-based polymers is an interesting strategy to produce PEMs with appreciable cell performance. However, I believe it would be useful to comment or compare the performance of SI-branched PPBP membranes to other viable polymer alternatives based PEMFCs demonstrating similar behavior, such as sPSU-LDH (C. Simari et al. Journal of Power Sources 471 (2020) 228440), SGO-sPEEK (K. Scott et al. RSC Adv., 2014, 4, 617); PBI_SiO2 (Y. Devrim, H. Devrim, and I. Eroglu, “Polybenzimidazole/SiO2 hybrid membranes for high temperature proton exchange membrane fuel cells,” Int. J. Hydrogen Energy, vol. 41, no. 23, pp. 10044–10052, 2016).
- Please go through the manuscript to eliminate the typos.
Author Response
In PEMFCs, the mechanical properties are also important because the membranes must tolerate severe mechanical solicitations. Please show the stress-strain tests on the proposed membranes.
Response: The mechanical properties are the foremost parameter of membrane for PEMFCs. Therefore, we have included Young’s Modulus values as the evidence of mechanical properties of the synthesized membranes in Table-1.
The development of sulfonimide-based polymers is an interesting strategy to produce PEMs with appreciable cell performance. However, I believe it would be useful to comment or compare the performance of SI-branched PPBP membranes to other viable polymer alternatives based PEMFCs demonstrating similar behavior, such as sPSU-LDH (C. Simari et al. Journal of Power Sources 471 (2020) 228440), SGO-sPEEK (K. Scott et al. RSC Adv., 2014, 4, 617); PBI_SiO2 (Y. Devrim, H. Devrim, and I. Eroglu, “Polybenzimidazole/SiO2 hybrid membranes for high-temperature proton exchange membrane fuel cells,” Int. J. Hydrogen Energy, vol. 41, no. 23, pp. 10044–10052, 2016).
Response: we have mentioned the performance of the above-mentioned polymer alternatives in the introduction part (marked as yellow) as you suggested.
Please go through the manuscript to eliminate the typos.
Response: We have thoroughly checked the manuscript again to eliminate the typos.
Reviewer 2 Report
The branched sulfonyl imide-based proton exchange polymer membranes from poly(phenylenebenzopheneone)s for fuel cells application is shown in this paper.
The vital constituent of the fuel cells is PEM (mainly Nafion). Although the Nafion characterize by the good chemical and oxidative stability but also characterize by high manufacturing cost, harsh chemical modifications, and high fuel permeability. Therefore, the search of the new PEMs is very important. Especially in the era of fast development of fuel cells and hydrogen technology. The paper shows that the analysed membranes improved the proton conductivity compare to Nafion 117. Thermogravimetric analysis and Fenton’s test study have ensured the significant thermal and chemical properties of the analysed membranes.
The document is characterized by a adequate cross-section of references. The references are well-matched to the subject of the document.
It is very interesting paper study of proton exchange polymer membranes for fuel cells application. Especially, in compare the results of research of analysed membranes with Nafion 117.
I believe this paper doesn't need any further revision.
But it should be good to extend and rewrite the conclusions for better understanding of the research results.
Author Response
The branched sulfonyl imide-based proton exchange polymer membranes from poly(phenylenebenzopheneone)s for fuel cells application is shown in this paper.
The vital constituent of the fuel cells is PEM (mainly Nafion). Although the Nafion characterize by the good chemical and oxidative stability but also characterize by high manufacturing cost, harsh chemical modifications, and high fuel permeability. Therefore, the search of the new PEMs is very important. Especially in the era of fast development of fuel cells and hydrogen technology. The paper shows that the analyzed membranes improved the proton conductivity compare to Nafion 117. Thermogravimetric analysis and Fenton’s test study have ensured the significant thermal and chemical properties of the analyzed membranes. The document is characterized by an adequate cross-section of references. The references are well-matched to the subject of the document.
It is a very interesting paper study of proton exchange polymer membranes for fuel cells application. Especially, in comparing the results of research of analyzed membranes with Nafion 117.
I believe this paper doesn't need any further revision.
But it should be good to extend and rewrite the conclusions for a better understanding of the research results.
Response: We have rewritten the conclusion of the manuscript to understand the results of the research.
Reviewer 3 Report
The authors prepared a series of branched sulfonimide-based proton exchange polymer membranes from poly(phenylenebenzopheneone)s for fuel cells application. This paper has the potential to be accepted, but some important points have to be clarified or fixed before we can proceed and positive action can be taken.
- Why did the authors choose fluorosulfonimide acid functional groups as an ion-conducting source? I suggest you include this consideration in the introduction.
- Why did the authors place the preparation and characterization of membranes in supplementary material? I suggest the author put them in the main article instead, since these are important.
- Did the authors follow formulas 2 and 6 from supplementary materials for measuring IEC and conductivity? If yes, then the word “usually” is not necessary here.
- The curves in Figure 2 overlap each other below 1500 cm-1. I suggest putting more space for each other.
- Please include error bars in Figures 3 and 4.
- The term should be water uptake not water content in Table 1.
- Did the authors measure the tensile strength? I found in lines 73-76 of supplementary materials wrote about tensile strength but I didn’t find the results.
Author Response
The authors prepared a series of branched sulfonimide-based proton exchange polymer membranes from poly(phenylenebenzopheneone)s for fuel cells application. This paper has the potential to be accepted, but some important points have to be clarified or fixed before we can proceed and positive action can be taken.
- Why did the authors choose fluorosulfonimide acid functional groups as an ion-conducting source? I suggest you include this consideration in the introduction.
Response: The causes of choosing the fluorosulfonimide acid groups as ion-conduction sources have been included in the introduction part of the revised manuscript. (Marked as yellow)
- Why did the authors place the preparation and characterization of membranes in supplementary material? I suggest the author put them in the main article instead since these are important.
Response: According to your suggestion the membranes preparation, characterizations, and measurements of the properties have been included in the main manuscript. (Marked yellow)
- Did the authors follow formulas 2 and 6 from supplementary materials for measuring IEC and conductivity? If yes, then the word “usually” is not necessary here.
Response: Yes. To measure the IEC and conductivity we have used formulas 2 and 6. Therefore, the term “usually” has been removed from the manuscript.
- The curves in Figure 2 overlap each other below 1500 cm-1. I suggest putting more space for each other.
Response: The spaces among the curves in figure 2 have been extended to avoid overlapping in the region 1500 cm-1.
- Please include error bars in Figures 3 and 4.
Response: Error bars have been included the Figures 3 and 4.
- The term should be water uptake not water content in Table 1.
Response: The term “water content” has been replaced by “ water uptake”.
- Did the authors measure the tensile strength? I found in lines 73-76 of supplementary materials wrote about tensile strength but I didn’t find the results.
Response: The tensile strength of the synthesized membranes has been measured in terms of Young’s Modulus and included in the table-1. (Marked yellow)
Reviewer 4 Report
In this study, some PEMs based on branched polyaromatics with sulfonic acid groups were prepared by using 1,3,5-trichlorobenzene as branching agent. The structure, morphology and properties of the membranes were characterized. I recommend its publication with the following modifications.
- For better understanding, please clearly indicate the name of each product in Scheme
- Is there any difference among the SI-PBP, SI-branched PBP (Figure 4) and SI-branched PPBP polymers. And please unify the name.
- If possible, please provide the photos of the membranes, and also give the shapes, thickness and color of the membranes.
- How about the mechanical properties of the membranes?
- Some typical references about side-chain-acid polyaromatics for highly stable PEMs can be cited (e.g. Macromolecules 2007, 40(6): 1934-1944; Journal of Power Sources 2008,185(2):899-903.
- Please confirm the structure of 1,3,5-trichlorobenzene in Scheme 1.
- Please check the sentence of “The branched SI-branched PPBP… …”.
- 1H NMR results are not clear to support the structure, and the main peaks are overlapped.
Author Response
In this study, some PEMs based on branched polyaromatics with sulfonic acid groups were prepared by using 1,3,5-trichlorobenzene as branching agent. The structure, morphology and properties of the membranes were characterized. I recommend its publication with the following modifications.
- For better understanding, please clearly indicate the name of each product in Scheme
Response: The name of each membrane has been specifically mentioned in the scheme.
- Is there any difference among the SI-PBP, SI-branched PBP (Figure 4) and SI-branched PPBP polymers. And please unify the name.
Response: Actually, it’s a typing mistake. It should be SI-branched PPBP. Therefore, all the names have been unified in the manuscript.
- If possible, please provide the photos of the membranes, and also give the shapes, thickness and color of the membranes.
Response: The shapes and thickness of the membranes have been included in table-1 of the manuscript. And the images of the synthesized membranes (Branched PPBP, S-branched PPBP, and SI-branched) have been included in Figure 7.
- How about the mechanical properties of the membranes?
Response: The mechanical properties of the synthesized membranes have been measured in terms of Young’s Modulus and mentioned in Table-1.
- Some typical references about side-chain-acid polyaromatics for highly stable PEMs can be cited (e.g. Macromolecules 2007, 40(6): 1934-1944; Journal of Power Sources 2008,185(2):899-903.
Response: The above-mentioned references have been cited in the manuscript. (Ref. no 11 & 19)
- Please confirm the structure of 1,3,5-trichlorobenzene in Scheme 1.
Response: The structure of the 1,3,5-trichlorobenzene has been corrected in Scheme 1.
- Please check the sentence of “The branched SI-branched PPBP… …”.
Response: The mentioned sentence has been corrected in section 3.2 of the manuscript. (Marked yellow)
- 1H NMR results are not clear to support the structure, and the main peaks are overlapped.
Response: As you are well known about the shifting of the N-H proton in 1H-NMR. Usually, in polymer, the N-H peak can be appeared in between 5~8 ppm and sometimes overlapped with the main aromatic ring peaks or remaining -SO3H groups bonded with water of the polymer. Sometimes the N-H proton peaks may also be absent. Therefore, 19F-NMR peaks for fluorosulfonimide can be supportive evidence of the structures. The fluorine peak for fluorosulfonime appeared at 139.449 ppm of the (Figure 1d) synthesized SI-branched PPBP polymer membranes.
Round 2
Reviewer 1 Report
The authors reply was very exhaustive.
The manuscript can be accepted in the present form